# Anti-Scale Performance and Mechanism of Valonia Tannin Extract for Calcium Carbonate in Circulating Cooling Water System

Zhenbo He [1], Li Zhang [1,\*] , Lihong Wang [1], Qiang Zhang [2] and Lingyu Luan [1,\*]

1    Shandong Analysis and Test Centre, Qilu University of Technology (Shandong Academy of Sciences), Jinan 250014, China; 10431201072@stu.qlu.edu.cn (Z.H.)
2    Center for Soil Pollution Control of Shandong, Shandong Department of Ecological Environment, Jinan 250012, China
\*    Correspondence: zlsatc@qlu.edu.cn (L.Z.); sdlly916@126.com (L.L.)

**Abstract:** Natural-polymer-based antiscalants for various calcium scales have recently received significant attention due to their prominent structural features, such as hydroxyl, amino, and organic acids, as well as their environmental friendliness and widespread availability. In this study, a novel green antiscalant, namely modified valonia tannin extract (MVTE), was synthesized using valonia tannin extract (VTE), itaconic acid (IA), and 2-acrylamido-2-methylpropanesulfonic acid (AMPS). The structure of MVTE was characterized by Fourier transform infrared spectroscopy (FT-IR). The crystal morphology, structure, and surface elementary composition of $CaCO_3$ were analyzed using scanning electron microscopy (SEM), X-ray diffraction (XRD), and X-ray photoelectron spectroscopy (XPS), respectively. Results indicate that MVTE with the best anti-scale performance is prepared when the valonia dosage is 2.5 g, the initiator dosage is 6 wt.%, the reaction temperature is 75 °C, and the reaction time is 3.5 h. Moreover, MVTE shows significantly improved resistance to temperature and alkalinity compared to VE. Results from SEM, XRD, and XPS demonstrate that MVTE can interfere with the regular growth of $CaCO_3$ crystals through chelation, dispersion, and lattice distortion. This effect results in the generation of vaterite, which inhibits the deposition of $CaCO_3$. Meanwhile, the molecular dynamics (MD) simulation was employed to further explore the anti-scale mechanisms at an atomistic level. The results illustrate that interaction energies originate from ionic and hydrogen bonds between MVTE and calcite, which ultimately improve the anti-scale performance of MVTE. In conclusion, MVTE can be an excellent antiscalant in circulating cooling water systems.

**Keywords:** antiscalant; calcium carbonate; valonia tannin extract; anti-scale mechanism; molecular dynamics simulation





## 1. Introduction

The demand for water has significantly increased due to the growing population and expanding industries, aggravating the existing water shortage crisis [1]. To alleviate the shortage of water resources, circulating cooling water systems are widely employed in industry. However, the continuous use of the cooling water can inevitably increase the concentration of scale-forming ions (i.e., $Ca^{2+}$, $HCO_3^-$, $CO_3^{2-}$, and $SO_4^{2-}$), leading to the deposition of inorganic calcium scales (i.e., $CaCO_3$ and $CaSO_4$) on the surface of the equipment pipeline, which will result in lower heating transfer efficiency and higher operation and maintenance costs [2]. Additionally, the adsorption of calcium scales on the surface of machinery can corrode transportation equipment and even shorten its service life [3,4]. Therefore, it is urgent to find effective methods to prevent or eliminate calcium scales.

Currently, adding chemical antiscalants to the circulating cooling water system is one of the most economical and effective methods [5,6]. The functions of antiscalants are to

inhibit the formation of crystal nuclei or crystals, disrupt the growth of crystals, and thus achieve excellent anti-scale performance [7–10]. Phosphonic-containing antiscalants, such as amino trimethylenephosphonic acid (ATMP) and 2-phosphonobutane-1,2,4-tricarboxylic acid (PBTCA), show outstanding inhibition abilities to prevent the formation of $CaCO_3$ scales [11,12]. However, the use of phosphorus-containing antiscalants can lead to the eutrophication of water bodies and the destruction of the ecological environment [13]. With increasing environmental concerns and discharge limitations, the development and application of new phosphorus-free and low-toxic green antiscalants are urgently necessary [14].

Polyaspartic acid (PASP) is a green polymer material that has great development prospects owing to its phosphorus-free composition and biodegradability [15,16]. PASP can function as an antiscalant due to its chelation and dispersion effects, attributed to the carboxyl group in its structure [14,17,18]. However, the anti-scale performance of PASP markedly decreased at high temperatures [19] or high $Ca^{2+}$ concentrations in solution [1], severely restricting its use. Previous studies have reported that polymerization is an effective method to produce multi-functional antiscalants that exhibit good anti-scale efficiency in extreme environments of high salinity, high alkalinity, and high temperature [20–22]. Zhang et al. [1] synthesized a PASP/urea copolymer using polysuccinimide (PSI) and urea as raw materials. The structure contains hydroxyl, carboxyl, and amide groups, which results in excellent anti-scale efficiency for $CaCO_3$, with 93% at a concentration of 10 mg/L. Zhao et al. [23] used PASP and aminomethanesulfonic acid (ASA) to prepare an effective antiscalant (PASP/ASA) by introducing sulfonic groups. The findings demonstrate that PASP/ASA shows superior anti-scale performance and high-temperature resistance when compared to PASP. As another commonly used typical green antiscalant, polyepoxysuccinic acid (PESA) contains carboxyl groups in its structure, which has a good anti-scale performance on the formation of calcium scales, but it has the disadvantages of low calcium tolerance and temperature resistance [24]. To compensate for these shortcomings, Yan et al. [25] synthesized a new type of phosphorus-free antiscalant containing carboxyl, sulfonic, and amide groups. The results show that it exhibits better resistance to $Ca^{2+}$ and temperature, as well as higher anti-scale efficiency, compared to PESA. Therefore, introducing carboxyl, sulfonic, and amide groups through polymerization is an effective method to prepare multi-functional green antiscalants with temperature and $Ca^{2+}$ resistance, as well as excellent anti-scale efficiency in circulating cooling water systems. Free radical polymerization is a common polymerization reaction [20,26,27]. Cui et al. [21] used maleic anhydride (MA), acrylic acid (AA), and 2-acrylamido-2-methylpropanesulfonic acid (AMPS) to prepare a copolymer called P(MA-AA-AMPS). The copolymer contains both sulfonic and amide groups, which greatly improves its chelating ability for $Ca^{2+}$ ions and its resistance to high temperatures and salinity.

Additionally, the development of sustainable green antiscalants is becoming the focus of researchers in academia and industry. Natural plant extracts with abundant hydroxyl, carboxyl, and amino functional groups (such as chitosan, inulin, starch, etc.) are another type of potential green antiscalant candidate [28–30]. Tannin extract (TE) is a complex secondary metabolite of higher plants, second only to lignin in biomass, and is widely distributed in plants (such as bark, fruit, leaves, roots, etc.) [31,32]. They can be classified into hydrolysable tannins and condensed tannins based on their composition [33]. Valonia tannin extract (VTE) is one of the tannin extracts, and the structure is shown in Figure S1a. VTE can be obtained through the process of extraction, filtration, washing, and drying of acorn powder [34]. It is a typical hydrolysable tannin, which is easily hydrolyzed to gallic acid and other phenolic acids in water [35]. Although there are other phenolic acids present in the hydrolysate, their relative content is very small. Therefore, gallic acid (Figure S1b) is commonly used as the smallest structure to represent VTE [35–37]. VTE contains various active groups, including phenol hydroxyl and carboxyl groups, that exhibit chelation and adsorption abilities with metal ions [31]. It is extensively used as a vegetable tanning agent [38], astringent agent, anticancer drug [38], corrosion inhibitor, and mineral processing inhibitor [39,40]. However, there have been few studies on using VTE

as an antiscalant. The hydroxyl groups in VTE are highly active [33], which enables VTE to capture the free radicals decomposed by the initiator, transfer the active site to the benzene ring [41], and increase the possibility of free radical polymerization. Therefore, VTE can be regarded as a potentially sustainable starting reagent to synthesize an antiscalant.

In this study, a novel green antiscalant called modified valonia tannin extract (MVTE) was prepared using valonia tannin extract (VTE), itaconic acid (IA), and 2-acrylamido-2-methylpropanesulfonic acid (AMPS) as starting reagents based on the principle of free radical polymerization. In an attempt to synthesize an excellent antiscalant, the influences of valonia dosage, initiator dosage, reaction temperature, and reaction time on anti-scale performance were investigated. The structure, molecular weight, and elemental composition of antiscalant were analyzed by Fourier transform infrared spectroscopy (FT-IR), an elemental analyzer, and gel permeation chromatography (GPC), respectively. Moreover, the anti-scale efficiency of the synthesized antiscalant for $CaCO_3$ and $Ca_3(PO_4)_2$ was estimated using a static scale inhibition experiment. Scanning electron microscopy (SEM), X-ray diffraction (XRD), and X-ray photoelectron spectroscopy (XPS) were used to explore the anti-scale mechanisms. Additionally, molecular dynamics (MD) simulation was employed to further investigate the interaction between the antiscalant and $CaCO_3$ crystal at an atomistic level [42].

## 2. Materials and Methods

### 2.1. Materials

IA and AMPS were purchased from Shanghai Maclin Biochemical Technology Co., Ltd., Shanghai, China. Bayberry, larch, black wattle, quebracho, chestnut, and valonia tannin extracts were purchased from Fujian Youqiyi Pharmaceutical Trading Co., Ltd., Fuzhou, China, and their components are shown in Table S1. Hydrochloric acid, potassium hydroxide, sodium bicarbonate, calcium chloride, potassium dihydrogen, phosphate sodium tetraborate decahydrate, ammonium persulfate, and potassium persulfate (KPS) were purchased from Sinopharm Chemical Reagent Co., Ltd., Shanghai, China. Additionally, EDTA was purchased from Tianjin Hengxing Chemical Preparation Co., Ltd., Tianjin, China. A calconcarboxylic acid indicator for scale inhibition was provided by Shanghai Aladdin Biochemical Technology Co., Ltd., Shanghai, China. Ultra-pure water was used for all experiments.

### 2.2. The Preparation Method of MVTE

The preparation process of MVTE is shown in Figure 1. Briefly, 8.5 g IA and 26.5 g AMPS were sequentially added to a four-necked round-bottom flask containing a thermometer, a magnetic stirrer, and a condensation reflux device. Approximately 40 g of ultra-pure water was then added, and the mixture was thoroughly stirred. A certain amount of valonia tannin extract was added to the flask and stirred until the reaction monomer was completely dissolved. The temperature was then raised to 75 °C. The potassium persulfate solution was slowly added to the flask, with a drop time of about 0.5 h. The reaction was maintained at 75 °C for 3.5 h to obtain a reddish-brown viscous liquid. The entire reaction was carried out under a nitrogen atmosphere until the experiment was completed. The synthesis mechanism of the MVTE is shown in Figure 2.



**Figure 1.** The schematic diagram of preparation process of MVTE.

**Figure 2.** The synthesis mechanism of MVTE.

### 2.3. Characterization

The structure was characterized using the Fourier transform infrared spectrometer (FT-IR, Thermo Scientific Nicolet iS20, America) at wavelengths ranging from 400 cm$^{-1}$ to 4000 cm$^{-1}$. The contents of C, H, N, and S of MVTE were analyzed on an elemental analyzer (Vario Elcube, Germany) [25], and the results are listed in Table S2. The molecular weight of MVTE was determined by means of gel permeation chromatography (GPC, calibrated with polyethylene glycol standards), and the results are listed in Table S3. At a flow rate of 1.0 mL/min, water was used as the mobile phase. $^{13}$C-NMR was recorded using an AVANCE HD 400 MHz NMR spectrometer (Bruker, Germany) with D$_2$O as a solvent. The surface morphology of CaCO$_3$ crystals in the absence and presence of 35 mg/L MVTE was observed using the scanning electron microscope (Zeiss SUPRA55, Germany) at an

acceleration voltage of 5 kV. X-ray diffractometer (PANalytical EMPYREAN, Holland) with Cu K$\alpha$ ($\lambda$ = 0.154 nm) radiation was conducted to study the crystal structures of $CaCO_3$ scales at a voltage of 40 kV and a current of 40 mA [43]. Moreover, the chemical states of various elements on the surface of $CaCO_3$ crystals obtained with and without 35 mg/L MVTE were determined via X-ray photoelectron spectroscopy (Thermo Scientific K-Alpha, America; h$\upsilon$ = 1486.6 eV), with contaminated carbon (C 1s = 284.8 eV) as the reference.

### 2.4. The Static Scale Inhibition Tests

The static scale inhibition tests were applied according to the national standard of P.R. China (GB/T16632-2008) [44]. The prepared calcium chloride and sodium bicarbonate test solutions were added to the 500 mL conical flask until $Ca^{2+}$ = 240 mg/L and $HCO_3^-$ = 720 mg/L, respectively [45]. The 0.01 mol/L borax buffer solution was slowly added before the required test antiscalant solution was added. At this time, a dilute hydrochloric acid or borax solution was used to adjust the pH of the solution to 9. The sample solution was then placed in a constant-temperature water bath at 80 °C for 10 h. After heating, the sample was allowed to cool to room temperature and filtered through medium-speed quantitative filter paper. Next, 20 mL of the supernatant was added to a 250 mL conical flask and diluted with pure water to a total volume of about 80 mL. Then, 5 mL of a 200 g/L potassium hydroxide solution was slowly added. Finally, the concentration of $Ca^{2+}$ in the solution was titrated via 0.01 mol/L EDTA. The anti-scale efficiency was then calculated based on the remaining $Ca^{2+}$ concentration in the solution using formula (1):

$$\eta(\%) = \frac{C_2 - C_0}{C_1 - C_0} \times 100 \tag{1}$$

where $\eta$ is the anti-scale efficiency, $C_1$ and $C_2$ are the concentrations of $Ca^{2+}$ in the absence and presence of antiscalant, respectively, and $C_0$ is the concentration of $Ca^{2+}$ in the test solution before reaction.

Calcium phosphate precipitation experiments were also carried out, similar to calcium carbonate precipitation. The procedure for calcium phosphate prepared solutions was changed into solutions of $Ca^{2+}$ of 250 mg/L and $PO_4^{3-}$ of 5 mg/L according to the national standard of P.R. China (GB/T 15452-2009) [46].

### 2.5. The Single Factor Method

The dosage of valonia (g), the dosage of initiator (wt.%), reaction temperature (°C), and reaction time (h) were found to significantly influence the molecular weight and spatial structure of the polymer, thereby directly impacting the anti-scale efficiency of the antiscalant [26]. Therefore, the effects of different factors on the anti-scale performance of the synthesized MVTE were investigated using the single-factor method and static scale inhibition experiments. During the optimization tests, a test solution with pH = 8 was used in all static scale inhibition experiments, and it contained 240 mg/L $Ca^{2+}$ and 720 mg/L $HCO_3^-$.

### 2.6. Molecular Dynamics Simulation

Molecular dynamics (MD) simulation was used to further investigate the mechanisms of interaction between MVTE and $CaCO_3$ crystals. This approach provided insights into the microscopic interactions between molecules at an atomistic level and helped to interpret the experimental results [47,48]. In this study, MD simulation was performed using the Forcite module of Materials Studio (MS) software [49].

#### 2.6.1. Simulation Parameters and Force Field

The COMPASS force field was an ab initio computational force field that was suitable for inorganic and organic molecules [42]. It had been successfully employed in studying the interaction between various molecules and calcite [50,51]. Therefore, the COMPASS force field used in this study was appropriate for meeting the computational requirements.

All MD simulations were conducted using both the canonical ensemble (NVT) and the Berendsen thermostat with medium convergence accuracy at 353 K [52,53]. The non-bond interaction of the system was calculated according to a cut-off radius of 1.25 nm [54]. To ensure that the initial structure reached thermodynamic equilibrium in sufficient time, a time step of 1 fs and a total simulation time of 300 ps were used. Trajectories were recorded every 5000 steps during the simulation. These parameters were essential in achieving thermal equilibrium and accurately capturing the molecular dynamics of the system.

2.6.2. Model Construction

In this study, the Visualizer module in MS was applied to establish molecular models [20]. Considering computing power, the structural unit of the polymer was selected for MD simulation [55,56]. Three random molecular structures of MVTE were generated, as shown in Figure S2. These structures were optimized using the molecular mechanics method (MM) to determine the minimum energy [42]. Their energies were −437.11, −424.88, and −420.26 kcal/mol, respectively. It was found that the energy of the structure shown in Figure S3a was the smallest; thus, this structure was selected for further study in regard to its interaction with calcite crystals.

The calcite cell belongs to a tripartite system with space group R3(-)C, and its cell parameters are as follows: a = b = 0.4991 nm, c = 1.7061 nm, $\alpha$ = $\beta$ = 90°, $\gamma$ = 120° [57]. Previous reports and experimental results indicated that the (104) surface was the main growth plane of calcite [58,59]. Hence, the interaction between antiscalants and the calcite (104) crystal surface was predominantly studied in this MD simulation. The interaction model between MVTE and calcite was constructed, as shown in Figure 3.

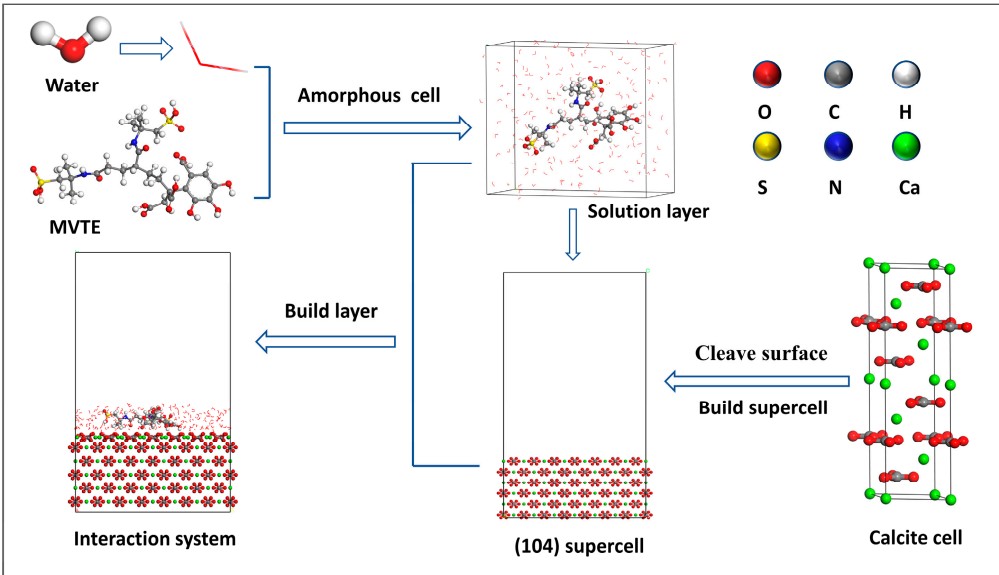

**Figure 3.** Construction of the interaction system of MVTE with calcite (104) surface.

To construct the interaction model of antiscalants with the calcite (104) surface, the calcite cell was first cut into a 2D (104) crystal plane consisting of six layers of atoms. The plane was then geometrically optimized. Next, a vacuum layer of about 3 nm was added above the crystal surface to obtain 4 × 4 3D periodic supercells (3.238 nm × 3.493 nm × 4.673 nm). The atoms within the bottom five layers of the supercell were constrained using coordinate constraints, while the top layer was left free before the entire supercell was optimized. Similarly, the corresponding water and organic molecules were also optimized and subsequently randomly added to the amorphous cell, called the solution layer. This solution layer consisted of one antiscalant molecule and two hundred water molecules [60] and was placed on the (104) surface at a distance of approximately 0.3 nm. To eliminate the influence of periodic boundary conditions, a vacuum layer with a thickness of 2.5 nm was

added above the solution layer along the Z-axis (c) direction [42]. Subsequently, geometry optimization and MD simulation were performed for the whole system. After reaching temperature and energy equilibrium, the interaction energy ($E_{inter}$) between the antiscalant and the crystal surface was calculated using Equation (2) [11]:

$$E_{inter} = E_{tot} - E_{surf+water} - E_{poly+water} + E_{water} \tag{2}$$

where $E_{tot}$ is the total energy of the calcite surface and solution layer, $E_{surf+water}$ is the total energy of the crystal surface and water molecules, $E_{poly+water}$ is the total energy of the solution layer, and $E_{water}$ is the total energy of water molecules. Binding energy ($E_{bin}$) is defined as the negative of the interaction energy ($E_{bin} = -E_{inter}$).

## 3. Results and Discussions

### 3.1. Determination of Optimum Synthesis Conditions of MVTE Antiscalant

#### 3.1.1. Selection of the Suitable Tannin Extract and Initiator

In this study, six common tannin extracts were used to synthesize effective antiscalants. The primary synthesis conditions were as follows: The dosage of the tannin extract was 3.5 g, the dosage of the initiator was 4% of the total mass, the reaction temperature was 80 °C, and the reaction time was 4 h. To select the most suitable raw material, static scale inhibition experiments were conducted to test the anti-scale performance of six different antiscalants synthesized from different tannin extracts against $CaCO_3$ scales. The results indicated that the modified valonia tannin extract (MVTE) synthesized by modifying valonia tannin extract (VTE) showed the best anti-scale performance (Figure S3). Therefore, VTE was chosen as the raw material for the following experiments. The $^{13}C$-NMR spectra of VTE are shown in Figure S4, and a detailed analysis is described in Text S1.

In order to select a more suitable initiator, two initiators, namely ammonium persulfate and potassium persulfate, were used to promote the synthesis of MTVE, respectively. The anti-scale performance of MVTE was tested, and the results are shown in Figure S5. It indicated that MVTE prepared using potassium persulfate as the initiator had better anti-scale performance than MVTE prepared using ammonium persulfate. Therefore, potassium persulfate was chosen as the initiator for the following experiments. Thus, VTE was selected as the raw material and potassium persulfate as the initiator in this study.

#### 3.1.2. The Effect of the Dosage of Initiator

The effect of the initiator dosage on the anti-scale performance was studied, and the results are shown in Figure 4a. It was evident that the anti-scale efficiency for $CaCO_3$ scales increased with the increasing initiator ratio, reaching the best anti-scale efficiency when the initiator dosage was 6%. A lower amount of initiators leads to insufficient production of free radicals, which results in incomplete copolymerization and the poor anti-scale performance of MVTE [60]. However, the anti-scale performance of the MVTE declined when the initiator amount exceeded 6%. This is because an excessive number of initiators leads to the generation of a large number of radicals, resulting in copolymerization occurring too fast and leading to a small molecular weight of the copolymer [4,20,61]. This makes the copolymer chelate fewer calcium ions, ultimately resulting in poorer anti-scale performance. As a result, the MVTE shows the best anti-scale efficiency when the KPS dosage is 6% of the total monomer mass.

#### 3.1.3. The Effect of the Dosage of Valonia

The effect of the valonia dosage on the anti-scale performance was evaluated, and the results are shown in Figure 4b. It was observed that the anti-scale efficiency improved with increasing the valonia dosage up to 2.5 g, reaching the best anti-scale efficiency at this dosage. However, the anti-scale performance decreased when the valonia dosage exceeded 2.5 g. The content of valonia is appropriately added to not only moderate the molecular weight of MVTE but also result in a reasonable number and distribution of functional

groups, which are critical to anti-scale performance [60,62]. Therefore, the dosage of 2.5 g of valonia is determined to be optimal for this study.

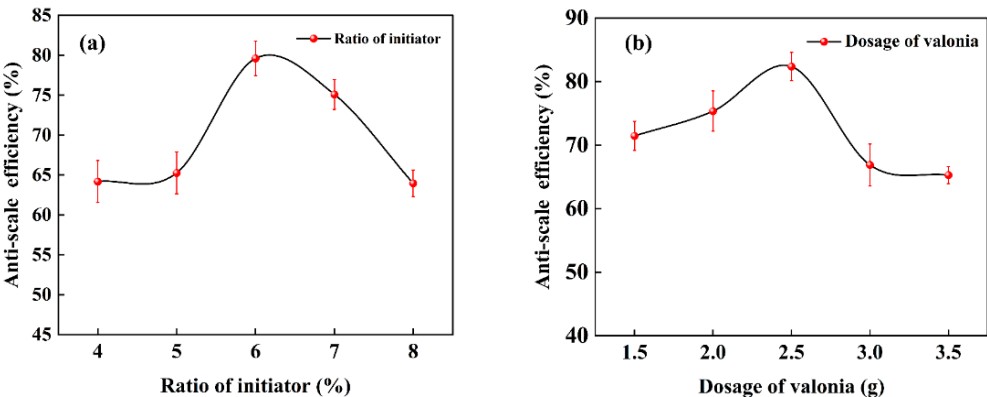

**Figure 4.** The effect of the dosage of initiator (**a**) and valonia (**b**) on the anti-scale performance.

### 3.1.4. The Effect of the Reaction Temperature

As shown in Figure 5a, the anti-scale performance gradually increased with the increasing reaction temperature, reaching its maximum at 75 °C, and decreased when the reaction temperature exceeded 75 °C. This is predominantly because the temperature has an effect on the initiator's decomposition and the polymerization reaction. The decomposition of initiators is an endothermic reaction that is hindered at lower temperatures, leading to an increase in copolymer chain length and resulting in steric effects [60]. In contrast, the radical polymerization reaction is an exothermic reaction, and excessively high temperatures can limit the reaction [63]. Hence, a temperature of 75 °C is chosen for this study.

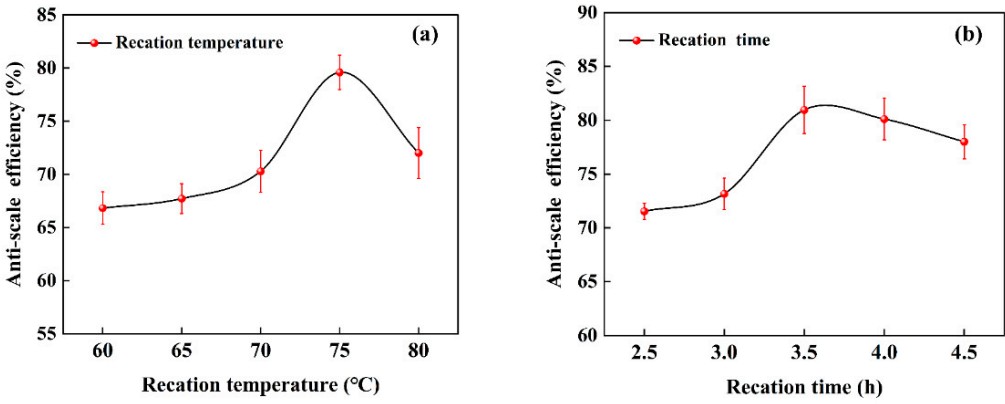

**Figure 5.** The effect of reaction temperature (**a**) and reaction time (**b**) on the anti-scale performance.

### 3.1.5. The Effect of the Reaction Time

The effect of the reaction time on the anti-scale performance was studied, and the results are displayed in Figure 5b. The analysis indicated that the anti-scale performance for $CaCO_3$ increased as the reaction time increased, achieving the highest anti-scale efficiency at 3.5 h. Subsequently, the anti-scale efficiency decreased with extended reaction time. This is mainly due to the effect of reaction time on the degree of polymerization. If the reaction time is too short, the reaction can be incomplete, leading to a low molecular weight of MVTE and a poor anti-scale effect. Alternatively, if the reaction time is too long, the molecular weight of MVTE can be too large, inhibiting the anti-scale effect of MVTE on $CaCO_3$ scales [62]. Therefore, the optimum reaction time is determined to be 3.5 h.

As a result, the optimal conditions for preparing MVTE were obtained using the single factor method, with the following conditions: Valonia dosage of 2.5 g, initiator dosage of 6.0%, reaction temperature of 75 °C, and 3.5 h reaction time. Under these conditions, the

MVTE exhibits excellent anti-scale performance for $CaCO_3$ scales. It is subsequently used to study the anti-scale performance in the next experiments.

### 3.2. The Characterizations of MVTE

Functional groups of VTE and MVTE were characterized using FT-IR, and the results are shown in Figure S6. It could be seen that there was a broad, strong stretching peak at around 3144 $cm^{-1}$, which was a combined effect of the stretching of O–H and N–H in the phenolic and amide groups [35,64]. In addition, the stretching peaks of unfunctionalized -$CH_3$ and -$CH_2$ were identified at 2990 and 2940 $cm^{-1}$, respectively [65]. The peak at 1731 $cm^{-1}$ might be assigned to the stretching vibration of C=O from aromatic esters or -COOH groups [66,67], while the absorption peak at 1661 $cm^{-1}$ was ascribed to the stretching vibration peak of C=O in the amide group [68]. The peaks at 1557, 1510, and 797 $cm^{-1}$ were inferred as the stretching vibrations of the $\upsilon$(C–C)/$\upsilon$(C=C) of the aromatic ring [37,62]. The absorption peak at 1398 $cm^{-1}$ was considered the characteristic absorption peak of $\alpha$-methylene in the carbonyl group. The peaks at 1308 and 1169 $cm^{-1}$ corresponded to the in-plane deformation vibration of the phenolic group $d_{ip}$(O-H) and the stretching vibrations of the phenolic group $\upsilon$(C–OH), respectively [37]. Additionally, the peaks at 1208 and 1039 $cm^{-1}$ were attributed to the stretching vibration of C–O from the phenol or -COOH group [69,70]. Subsequently, the peak at 523 $cm^{-1}$ was ascribed to the presence of the sulfonic acid group [71]. There were also two peaks at 1039 $cm^{-1}$ and 628 $cm^{-1}$, which were attributed to the stretching vibrations of S=O and S–C, respectively [20,21]. From the above absorption peaks of MVTE, it can be implied that VTE, AMPS, and IA participated in the reaction.

Furthermore, the element composition of MVTE was analyzed, and the corresponding results are listed in Table S2. The measured contents of C, N, and S in MVTE were found to be 37.26%, 4.20%, and 10.23%, respectively, which were in agreement with the theoretical values. This further suggested the successful introduction of sulfonic and amide groups into MVTE.

In addition, the results for weight average molecular weight ($M_w$) and number average molecular weight ($M_n$) of MVTE are presented in Table S3. The $M_w$ and $M_n$ were 4776 g/mol and 3915 g/mol, respectively. And MVTE contained approximately 7 IA and 14 AMPS. The polydispersity index (PDI = $M_w/M_n$) was 1.22, which indicated that the starting reagents were satisfactorily polymerized to produce a uniform polymer [25].

### 3.3. Analysis of Anti-Scale Performance of MVTE

#### 3.3.1. Effect of Different MVTE Concentrations against $CaCO_3$

The anti-scale performance of MVTE for the $CaCO_3$ precipitates was studied and compared with VE at various concentrations when the pH was 9, as shown in Figure 6a. It was observed that there was a significant improvement in the anti-scale efficiency of MVTE compared to VE, demonstrating that the addition of carboxyl, amide, and sulfonic groups played an important role in restricting the formation of $CaCO_3$ precipitates [46,70,71]. The anti-scale performance increased from 5 mg/L to 35 mg/L, reaching the maximum value of over 70% at an optimal concentration of 35 mg/L, which was comparable to the previous literature results and better than PASP under the same conditions [72]. One of the primary reasons for this improvement is the addition of functional groups that can chelate more $Ca^{2+}$ to directly prevent the generation of $CaCO_3$ crystals [73,74]. Another key factor is the ability of MVTE to adsorb on the surface of $CaCO_3$ crystals, leading to dispersion and lattice distortion [70]. Considering efficiency and practicality, a concentration of 35 mg/L MVTE is selected for use in the following studies.

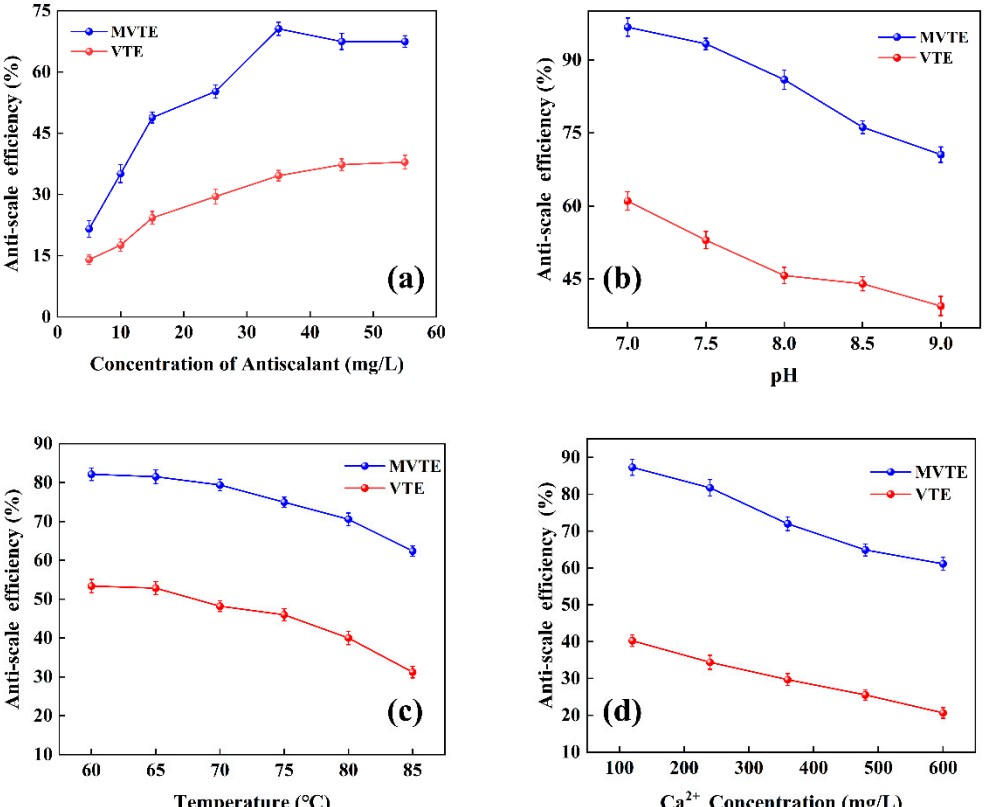

**Figure 6.** The impact of antiscalant concentration and water conditions on anti-scale performance: (**a**) the effect of different antiscalant concentrations at pH = 9 on anti-scale performance; (**b**) the effect of pH on anti-scale performance; (**c**) the effect of temperature at pH = 9 on anti-scale performance. (**d**) the effect of $Ca^{2+}$ concentration on anti-scale performance.

### 3.3.2. Effect of pH on the Anti-Scale Performance

The anti-scale performance under different solution pH values was demonstrated in Figure 6b at 80 °C. It was indicated that the anti-scale performance decreased as the pH of the solution increased from 7 to 9, but was still maintained above 70%. Notably, the anti-scale efficiency of MVTE reached up to approximately 98% at pH 7, indicating that MVTE exhibited better effects when used in low-alkalinity water systems. To explain this phenomenon, the following chemical equations were listed [11,62].

$$HCO_3^- \rightleftharpoons H^+ + CO_3^{2-} \tag{3}$$

$$HCO_3^- + H_2O \rightleftharpoons H_2CO_3 + OH^- \tag{4}$$

A previous report has shown that the pH of the solution is a key factor that affects the anti-scale efficiency of the antiscalant [72]. Chemical reactions (3) and (4) indicate that as pH increases, the hydrolysis of $HCO_3^-$ can be preferentially hindered, leading to increased ionization of $HCO_3^-$ and the induction of more $CO_3^{2-}$ to combine with $Ca^{2+}$ for the formation of $CaCO_3$ precipitations. This ultimately results in a poor anti-scale effect. However, after introducing the sulfonic and amide groups, the alkaline resistance of MVTE is significantly enhanced compared to VE.

### 3.3.3. Effect of Temperature on the Anti-Scale Performance

The influence of different temperatures on the anti-scale performance was investigated and plotted in Figure 6c while maintaining the pH value of 9 in the solution. As the temperature increased from 60 to 85 °C, the anti-scale performance gradually declined.

However, compared to VE, MVTE still exhibited better anti-scale efficiency at 80 °C, with the efficiency remaining above 70%. Firstly, at 80 °C, $HCO_3^-$ tends to dissociate into $CO_3^{2-}$ due to the presence of the borax buffer [75]. Moreover, the activities of $HCO_3^-$ and $Ca^{2+}$ increase at higher temperatures, resulting in an elevated effective collision between $HCO_3^-$ and $Ca^{2+}$ and thereby facilitating the formation of $CaCO_3$ scales [69]. However, the introduction of sulfonic and amide groups in MVTE enhances its high-temperature resistance compared to VE, which is crucial for the production of an effective antiscalant [21,76]. Therefore, MVTE can be effectively applied to circulating cooling water systems.

### 3.3.4. Effect of the Concentration of $Ca^{2+}$ on the Anti-Scale Performance

As shown in Figure 6d, it demonstrated that the anti-scale performance of MVTE against $CaCO_3$ deposits decreased as the concentration of $Ca^{2+}$ increased. At a $Ca^{2+}$ concentration of 120 mg/L, the anti-scale efficiency was 87.31%, and even when the $Ca^{2+}$ concentration reached 600 mg/L, the anti-scale efficiency was still above 61.01%. This is because as the concentration of $Ca^{2+}$ increases, excessive $Ca^{2+}$ can cause the accumulation of nuclei and crystals, which accelerates the formation of $CaCO_3$ deposits [19,21]. Compared to VE, MVTE shows good resistance to $Ca^{2+}$ due to the introduction of sulfonic and amide groups, which enhance its chelating and dispersing abilities [77].

### *3.4. Anti-Scale Mechanisms*

### 3.4.1. SEM Analysis of Scale Crystal Morphology

The changes in crystal morphology were analyzed using SEM to explore the anti-scale mechanism. As shown in Figure 7a–c, well-defined structures with cube and strip-like shapes were observed and found to be consistent with previous reports, identifying them as calcite and aragonite, respectively [78,79]. However, the crystal size, morphology, and structures of $CaCO_3$ crystals changed after the addition of the 35 mg/L MVTE (Figure 7d,e). This is because the carboxyl, amide, and sulfonic groups of MVTE have powerful affinities with $Ca^{2+}$, resulting in the retardation of nucleation and a reduction in the number of crystal nuclei [54,60,80]. Furthermore, MVTE can bind to the active growth positions of the $CaCO_3$ crystal surface through an electrostatic interaction [81], impeding the regular growth of the crystal and distorting the crystal lattice, ultimately resulting in the inhibition of scale formation [1,82]. As a result, $CaCO_3$ crystals become much rougher and looser, with mainly spherical particles of smaller size that are typical of the most thermodynamically unstable vaterite polymorph [19,75,83].

### 3.4.2. XRD Analysis of Scale Crystal Structure

For further $CaCO_3$ crystal characterization, the crystal structure in the absence and presence of 35 mg/L MVTE was characterized using XRD. As shown in Figure S7a, it could be seen that the diffraction peaks at 29.45°, 36.16°, 47.59°, and 48.42° were assigned to the (104), (110), (018), and (118) crystal planes of the calcite phase, respectively [84,85]. This indicated that the (104) plan was the main growth plane of calcite. Additionally, the characteristic peaks of aragonite were observed in the form of diffraction peaks at 26.30°, 42.96° and 45.94°, which corresponded to the (111), (122), and (221) crystal planes, respectively [11,19,86]. However, the presence of 35 mg/L MVTE (Figure S7b) resulted in the disappearance of the main peaks corresponding to the (111), (122), and (221) crystal planes of aragonite. At the same time, new diffraction peaks appeared at 24.92°, 27.08°, 32.87°, 43.90°, and 50.12°, which were attributed to the (110), (111), (112), and (114) crystal planes of vaterite, respectively [46,79,87]. Vaterite, being the most unstable form of $CaCO_3$ crystal, could be easily removed, as demonstrated by a previous report [88]. It is because the presence of carboxyl, amide, and sulfonic groups in the MVTE allows it to chelate with the $Ca^{2+}$ ions and occupy growth points on the scale surface, which hinders the nucleation and growth of calcium scale crystals and causes distortion of the crystal lattice [89–91]. Results of SEM and XRD indicate that MVTE exhibits the ability to retard crystal nucleation by binding to $Ca^{2+}$ and to disturb the growth of crystals by attaching to the active sites of the

crystal surface. As a result, the prepared MVTE exhibits excellent anti-scale performance on $CaCO_3$ precipitates.

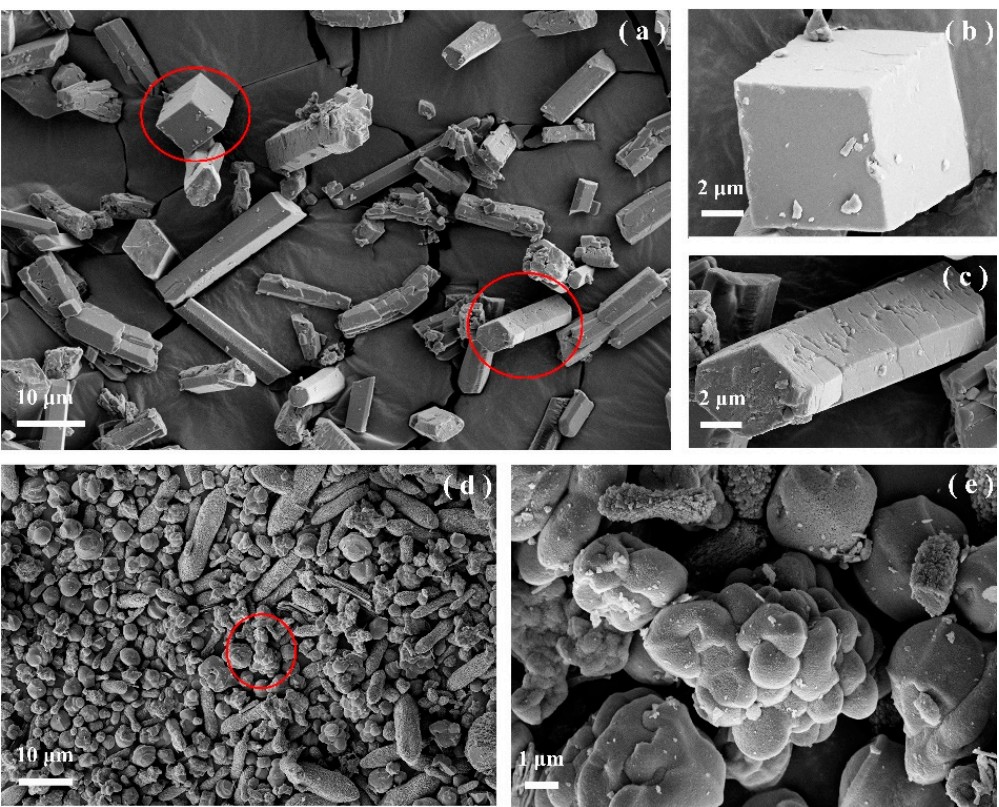

**Figure 7.** (**a**) SEM image of $CaCO_3$ without the presence of MVTE; (**b**) and (**c**) from the red cycle of (**a**), respectively; (**d**) SEM image of $CaCO_3$ with 35 mg/L MVTE; and (**e**) from the red cycle of (**d**).

### 3.4.3. XPS Analysis of Scale Crystal

To further gain insight from the anti-scale mechanism, XPS was employed to determine the surface elementary composition of the crystals in the absence and presence of 35 mg/L MVTE. As shown in Figure S8, the composition of $CaCO_3$ crystals with or without MVTE included C, O, Ca, and N (originating from MVTE). The C 1s peak (Figure S8a) was resolved into three component peaks, with the peak at 284.80, 285.94, and 289.60 eV corresponding to the C–H, C=O, and C–O, respectively [80,92]. After the addition of MVTE, a new peak (Figure S8d) emerged at 288.08 eV assigned to the N–C=O group from amide [93], which was possibly attributed to the adsorption of the MVTE on the $CaCO_3$ surface. The Ca 2p peaks (Figure S8b) exhibited two distinctive peaks at 347.34 and 350.92 eV, corresponding to Ca $2p_{3/2}$ and Ca $2p_{1/2}$ [94], respectively. However, after the addition of 35 mg/L MVTE, the Ca $2p_{3/2}$ and Ca $2p_{1/2}$ peaks shifted to 347.11 and 350.63 eV, respectively (Figure S8e). These shifts might be attributed to the functional groups of MVTE, such as -COOH, -OH, and -N–C=O, which could bind calcium ions and donate electrons, leading to an increase in the electron density of calcium ions [72]. The O 1s peaks (Figure S8c) exhibited a peak at 531.03 eV, which was ascribed to either $CaCO_3$ or C–O, while another peak was assigned -C=O from carboxyl or amide [93,95]. Additionally, the N 1s peak at 399.2 eV (Figure S8f) corresponded to the N–C from the amide of MVTE [93]. The XPS, SEM, and XRD results imply that the introduction of functional groups in MVTE can enhance the chelation with $Ca^{2+}$ ions and promote adsorption on the active sites of the $CaCO_3$ crystal surface, which leads to the lattice distortion of the $CaCO_3$ crystals and contributes to the excellent anti-scale performance of MVTE [86]. As a result, the anti-scale mechanisms of MVTE for $CaCO_3$ scales are presented in Figure 8.

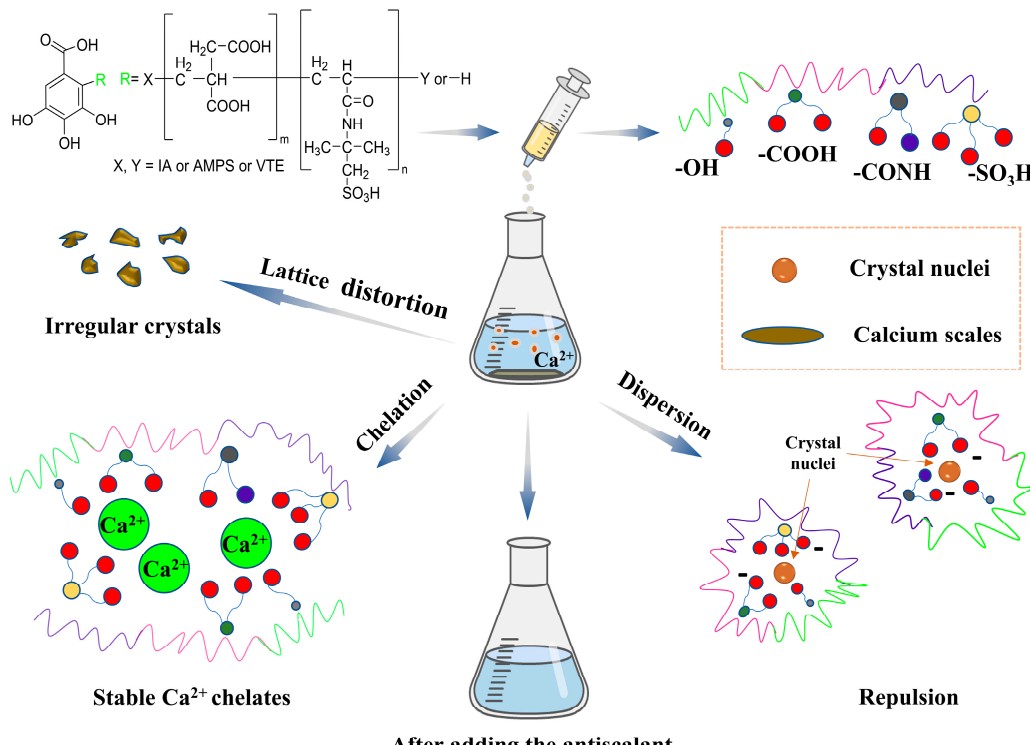

**Figure 8.** Schematic diagram presented the anti-scale mechanisms of MVTE for calcium carbonate scales.

### 3.5. Analysis of MD Simulation

#### 3.5.1. The Equilibrium Criteria of Interaction Systems

To ensure the reliability of the MD simulation results, it is important to confirm whether the system has reached equilibrium [42]. Equilibrium criteria were applied to determine the stability of the system, and the results are shown in Figure S9. The energy and temperature curves fluctuated within 10% during the 150~300 ps simulation time, indicating that the interaction system had reached equilibrium successfully [28]. The equilibrium models of VE, MVTE, and (104) crystal surfaces after MD simulation are illustrated in Figure S10.

#### 3.5.2. Binding Energy of Antiscalant on the Surface of Calcite

The binding energies of VE and MVTE on the (104) surface of calcite are listed in Table 1. It could be observed that the $E_{inter}$ energies between antiscalants and the $CaCO_3$ crystal surface were negative, indicating an exothermic reaction. This suggests that the antiscalant can spontaneously adsorb on the surface of $CaCO_3$ crystal [96], mainly due to the interaction between the $\pi_4^3$ delocalized bond in the carboxyl group and the $\pi_4^6$ delocalized bond of calcite [49]. Therefore, the polar functional groups in the antiscalant can adsorb onto the active sites on the surface of calcium scales, disrupting the regular growth of crystals. Additionally, the $E_{bin}$ value represented the strength of the interaction between the antiscalants and crystal surface, and the higher the value of $E_{bin}$, the better the anti-scale performance of the antiscalant [57]. It was evident that the $E_{bin}$ of MVTE was higher than that of VE, indicating that MVTE exhibited better anti-scale performance than VE. This is due to the introduction of carboxyl, amide, and sulfonic groups in MVTE, which can chelate with more $Ca^{2+}$, effectively hindering the formation of $CaCO_3$ crystals, leading to improved anti-scale performance compared to VE [56,97]. These findings are consistent with the results of the static scale inhibition experiments, further supporting the effectiveness of MVTE as an antiscalant.

**Table 1.** Interaction energies of antiscalant–calcite systems (kcal/mol).

| Surface | Antiscalant | $E_{tot}$ | $E_{sur+water}$ | $E_{poly+water}$ | $E_{water}$ | $E_{inter}$ | $E_{bin}$ |
|---|---|---|---|---|---|---|---|
| (104) | VTE | −46,637.28 | −46,575.47 | −787.18 | −759.86 | −34.49 | 34.49 |
| | MVTE | −47,010.13 | −46,460.63 | −1146.16 | −714.71 | −118.04 | 118.04 |

### 3.5.3. Deformation Energy of Antiscalant on the Surface of Calcite

The structure of the antiscalant was found to deform during its interaction with calcite, and the degree of deformation was determined by the magnitude of deformation energy ($E_{def}$), which was calculated using the following equation [42]:

$$E_{def} = E_{poly-bind} - E_{poly-free} \tag{5}$$

where $E_{poly-bind}$ and $E_{poly-free}$ refer to the single energy of the polymer after binding with the surface of the $CaCO_3$ crystal and the free state of the polymer, respectively.

The results of $E_{def}$ are listed in Table 2. For both VE and MVTE, the values of $E_{poly-bind}$ were positive, indicating that they underwent deformation on the crystal surface and required absorption energy. However, the absolute value of the non-bond energy ($\Delta_{Enon-bond}$) associated with the interaction between the antiscalant and crystal surface was much greater than $E_{def}$, illustrating its capability of overcoming its own deformation and adsorbing onto the crystal surface to support the normal growth of calcium scales [49]. Furthermore, the higher absolute value of $E_{def}$ indicates a greater degree of polymer deformation [49], leading to more stable adsorption on the surface of calcium scales and better anti-scale performance [50]. In the case of MVTE and VE, the $E_{def}$ of MVTE was greater than that of VE, indicating that MVTE had stronger deformation ability and a higher chance of interacting with the crystal surface, hence achieving better anti-scale performance, which was in agreement with the experimental results.

**Table 2.** The nonbonding interaction energies of antiscalant–calcite systems and the components of nonbonding interaction energies (kcal/mol).

| Surface | Antiscalant | $\Delta E_{non-bond}$ | $\Delta E_{electrostatic}$ | $\Delta E_{vdw}$ | $E_{def}$ |
|---|---|---|---|---|---|
| (104) | VTE | −47,240.41 | −51,737.61 | 4502.92 | 15.03 |
| | MVTE | −47,685.23 | −52,318,069 | 4640.26 | 58.49 |

### 3.5.4. Radial Distribution Function of Antiscalant with Calcite Surface

To further elucidate the relationship between the antiscalant and the calcium scale crystal, the radial distribution function (RDF) was utilized to analyze the trajectories obtained from the MD simulation. RDF represents the ratio between the regional density and the global density of another type of atom found within a certain range centered on one type of atom [47]. Generally, in the g(r)~r curve, the peak at r < 3.5 Å is primarily formed by chemical bonds and hydrogen bonds, while the peak at 3.5 Å < r < 5.0 Å mainly consists of Coulomb and van der Waals forces [98,99].

In this study, the g(r)~r curves were plotted for H atoms, O atoms in the antiscalant, and O atoms and Ca atoms on the (104) surface of calcite, as shown in Figure 9. The g(r)$_{O-Ca}$ curve of O atoms in the antiscalant and Ca ions in calcite exhibited a sharp peak value of around 2.40 Å that matched the length of the Ca–O electrovalent bond (2.39 Å) [49]. This indicates that chemisorption occurs between the negatively charged O atoms and the positively charged calcium ions through ionic bonds [52]. Moreover, the peak value of g(r)$_{O-Ca}$ of MVTE was larger than that of VE, agreeing with the results of binding energy. Additionally, a strong peak appeared around 1.4 Å in the curve of g(r)$_{H-O}$, slightly larger than the sum of the covalent radius (0.11 nm) of O and H atoms. This demonstrates the formation of hydrogen bonds between H atoms of the antiscalant and O atoms of calcite, which helps the antiscalant absorb onto the surface of crystals, improving the anti-scale

performance [100,101]. Therefore, the RDF results suggest that the interaction energy between MVTE and calcite mainly originates from ionic and hydrogen bonds.

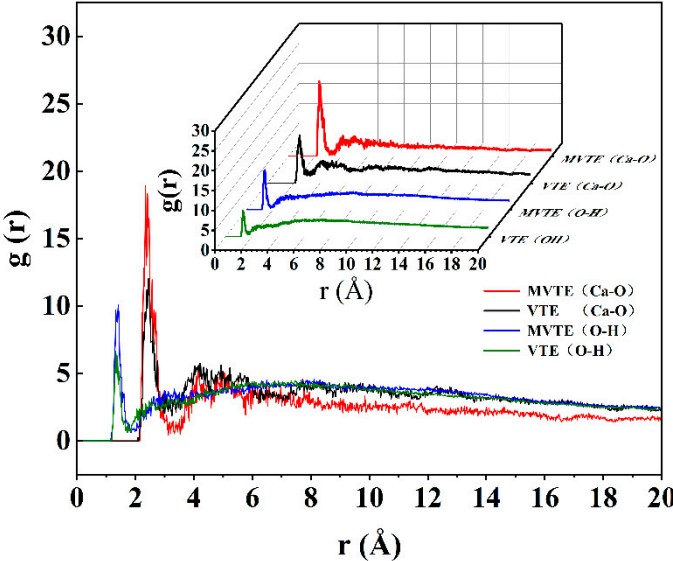

**Figure 9.** RDFs of VE, MVTE with calcite (104) surface.

Additionally, to gain intuitive insight into the interaction mechanisms between MVTE and the calcite surface, a detailed analysis of intermolecular interactions was conducted and presented in Figure 10. The analysis revealed that the distance between the O atoms of MVTE and the Ca atoms on the calcite surface ranged from 2.3 to 2.4 Å, while the distance between the H atoms of MVTE and the O atoms of the calcite surface ranged from 1.3 to 1.5 Å. Therefore, it can be inferred that the interactions between the MVTE and calcite surfaces primarily consist of ionic and hydrogen bonds, which is consistent with the results of RDF analysis.

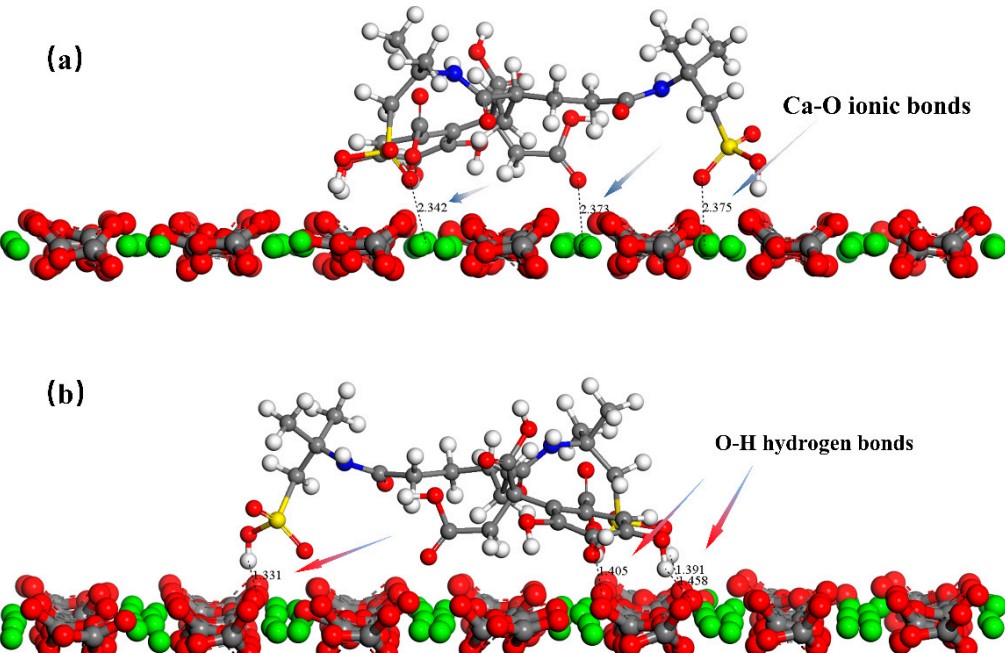

**Figure 10.** Distance between MVTE and calcite (104) surface: (**a**) between $Ca^{2+}$ of calcite and O atom of MVTE; (**b**) between O of calcite and H of MVTE.

### 3.6. The Anti-Scale Performance of MVTE on Calcium Phosphate

To expand the range of applications, the anti-scale performance of MVTE on $Ca_3(PO_4)_2$ was also investigated in this study. Under conditions of $C(Ca^{2+})$ = 250 mg/L, $C(PO_4^{3-})$ = 5 mg/L, pH = 9, and heating at 80 °C for 10 h, the anti-scale performance of different dosages of VE and MVTE was tested, with the results shown in Figure S11. At a concentration of 25 mg/L, the anti-scale efficiency of VE and MVTE on $Ca_3(PO_4)_2$ was 50.52% and 89.51%, respectively. In comparison to VE, MVTE demonstrated a significant improvement in anti-scale efficiency on $Ca_3(PO_4)_2$, illustrating the essential role played by the introduction of carboxyl, amide, and sulfonic groups [25]. These groups can enhance the chelating ability of $Ca^{2+}$, inhibiting the regular growth of calcium scales [21]. These findings suggest that MVTE shows excellent anti-scale performance on $Ca_3(PO_4)_2$, thereby emphasizing its potential as an antiscalant for $Ca_3(PO_4)_2$ in circulating cooling water systems.

## 4. Conclusions

In this study, optimal conditions for preparing MVTE were determined as follows: a valonia dosage of 2.5 g, an initiator dosage of 6% (ratio of the total mass of monomers), a reaction temperature of 75 °C, and a reaction time of 3.5 h. Results of FT-IR, GPC, and element composition analysis suggest that sulfonic and amide groups were successfully introduced into MVTE. At pH 7 and a temperature of 80 °C, MVTE shows an anti-scale efficiency of 98% for $CaCO_3$ scales at 35 mg/L. Furthermore, at a concentration of 25 mg/L, MVTE exhibits an anti-scale efficiency of 90.51% on $Ca_3(PO_4)_2$, which is superior to that of VE. Moreover, SEM, XRD, and XPS analyses indicated that MVTE can chelate calcium ions and adsorb onto active sites of calcium scale crystals, interfering with the regular growth of $CaCO_3$ scales by chelation, dispersion, and lattice distortion. Furthermore, MD simulation analyses demonstrate that MVTE molecules deform on the surface of calcium scales and release energy. Additionally, the analysis of RDFs reveals that interaction energies originate from ionic and hydrogen bonds between MVTE and calcite. This allows MVTE to combine with $CaCO_3$ crystals and hinder the regular growth of $CaCO_3$, resulting in crystalline lattice distortion. Compared to VE, MVTE shows significantly better tolerance to changes in temperature, alkalinity, and $Ca^{2+}$ concentration, attributed to the presence of carboxyl, sulfonic, and amide groups. These improvements are crucial for the application of MVTE in circulating cooling water systems.

**Supplementary Materials:** The following supporting information can be downloaded at: https://www.mdpi.com/article/10.3390/su15118811/s1, References [34,35,102] are cited in the Supplementary Materials.

**Author Contributions:** L.L. and L.Z. put forward the main research points; Z.H. and L.Z. completed manuscript writing and revision; L.Z., Z.H. and L.W. completed theoretical simulation; Z.H., L.L. and Q.Z. completed anti-scale performance research; L.Z., Z.H. and L.W. completed anti-scale mechanisms research; Z.H., L.L. and L.Z. revised grammar and expression. All authors have read and agreed to the published version of the manuscript.

**Funding:** This research was funded by National Natural Science Foundation of China (grant number 21902091), Shandong Province Natural Science Foundation (grant number ZR2019BB084), Shandong Province Natural Science Foundation (grant number 2022TSGC2276).

**Institutional Review Board Statement:** Not applicable.

**Informed Consent Statement:** Not applicable.

**Data Availability Statement:** No new data created.

**Acknowledgments:** This work was jointly supported by the Project of the National Natural Science Foundation of China (21902091) and the Shandong Province Natural Science Foundation (ZR2019BB084, 2022TSGC2276). We would like to express our gratitude to Yanbang Li from Kyoto University for his help in theoretical simulation.

**Conflicts of Interest:** The authors declare no conflict of interest.

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
