# Peer review of "Anti-Scale Performance and Mechanism of Valonia Tannin Extract for Calcium Carbonate in Circulating Cooling Water System"

_sustainability, doi:10.3390/su15118811_

Round 1
Reviewer 1 Report
Dear Authors,
Here are my comments and suggestions:
1. Please rewrite the first paragraph of the introduction.
2. Molecular dynamics (MD) simulation was used to further investigate the mechanisms of interaction between MVE and CaCO3 crys-170 tals.
3. In Fig. 13, can you indicate in (a) from where (b) and (c) were viewed? Do same for (d), (e), and (f). Importantly, I doubt if the magnifications are accurate. Please double check. Thanks.
4. Briefly explain how the scanning 133 electron microscope (Zeiss SUPRA55, Germany) was carried out. Do same for FT-IR and X-ray diffraction analysis (PANalytical 134 EMPYREAN, Holland). Indicate the voltage and radiation used. You can cite this paper: Agbabiaka, Okikiola Ganiu, et al. "BN-PVDF/rGO-PVDF Laminate Nanocomposites for Energy Storage Applications." Nanomaterials 12.24 (2022): 4492.
5. Move Figures 8, 14, and 15, to supporting document. Also, please reorganize all other figures.
6. Remove the graphical abstract. It looks somewhat the same as what is presented in the body.
The overall English quality of the writeup can be improved upon. There are paragraphs that need to be rewritten few of which have been highlighted in the comment section.
Author Response
Dear Experts:
Thanks for your comments and suggestions regarding our manuscript entitled “Anti-scale Performance and Mechanism of Valonia Tannin Extract for Calcium Carbonate in Circulating Cooling Water System” (2368780). Those comments and suggestions are valuable and very helpful. We have carefully read through the comments and made the necessary corrections. Revisions in the text are highlighted using red for additions and strikethrough for deletions.
I look forward to your reply.
Sincerely.
Zhenbo He.

Reviewer 2 Report
The comments are attached in the file.

Minor English checks are required in the manuscript.
Author Response

(The authors gave the same response as above.)

Reviewer 3 Report
Usage of natural extracts containing polysachharides and other compounds reach in carboxylic and phenolic groups is a popular direction in green chemistry approach for design of inhibitors of scale deposition. The approach suggested in this manuscript to enhance performance of the natural substances is based on introduction of new functional groups (sulfonic) and increase of molecular weight via grafting polymeric fragments under optimized reaction conditions. Although, the methodology of the investigation of anti-scale performance is adequate for this topic, the characterization of antiscalant is completely inappropriate. This fact eliminates impact of the reported research.
1. Why modified valonia extract (VE) is synthesized without mentioning Valonia and extraction? TE was purchased and used for synthesis of modified VE as a separated product but Figure 1 shows structural formula of Gallic acid, which is labelled as VE. Besides, the dosage of valonia (g) is mentioned in section 2.5. This is really misleading. What was used in the modification reaction and what was characterized for anti-scale properties in completely not understandable. Line 229: The authors write that “In this study, six common tannin extracts were used to synthesize effective antiscalants” but Figure S2 report results for the undefined extract. More explanations are required for structural difference between expected reaction products of bayberry and valonia. used to obtain data in Fig.5.The content and ratio of different polyphenolic acids in whole extracts can be variable, but this can affect composition of the final product. Thus, if the whole extract was used, it must be characterized.
2. There is no elemental composition, e.g. sulfur content in the MVE. Molecular weight of MVE is also not characterized, although it is assumed to affect performance. FTIR gives only qualitative information and it is not enough to understand characteristics of the products used in the experiments.
3. Conditions of the experiments must be indicated in Figure captures, e.g. pH shall be included to Figure 9 caption, concentration and pH to Figure 11 caption.
4. Many abbreviation is the introduction are given without full description, this must be checked and corrected.
The authors have to reconsider what they have used as a starting reagent, what they have obtained and investigated as modified product. If tannic acid was used, it is necessary to show that the same result can be obtained with the whole VE of the defined composition. Otherwise, there is no correlation between the title and the content of manuscript.
Only minor correction of the English is required.
Author Response

(The authors gave the same response as above.)

Round 2
Reviewer 1 Report
Please find my comments below.
1. With the increase in population and development of industries.....please correct.
2. Figures S3 to S11 are missing in the supporting document. Please double check.
3. Table S2 to S3 are missing as well.
The quality of English Language of the paper has been improved.
Author Response

(The authors gave the same response as above.)

Reviewer 3 Report
The manuscript was substantially revised and can be accepted for publication.
The English is Ok, minor corrections are required.
Author Response

(The authors gave the same response as above.)

Round 3
Reviewer 1 Report
Dear Authors,
The paper has been significantly improved upon. But, my concern is the first sentence of the introduction.
"With the increase of population and the expansion of industries, the demand for water has significantly increased, which further aggravates the already existing water shortage crisis"
May I suggest this instead-
The demand for water has significantly increased due to the growing population and expanding industries, aggravating the existing water shortage crisis.
The English quality of the paper is OK except for minor corrections.
Author Response

(The authors gave the same response as above.)
